# Epigenetic and Proteomic Biomarkers of Elevated Alcohol Use Predict Epigenetic Aging and Cell-Type variation Better Than Self-Report

**DOI:** 10.3390/genes13101888

**Published:** 2022-10-18

**Authors:** Steven R. H. Beach, Mei Ling Ong, Frederick X. Gibbons, Meg Gerrard, Man-Kit Lei, Kelsey Dawes, Robert A. Philibert

**Affiliations:** 1Department of Psychology, University of Georgia, Athens, GA 30602, USA; 2Center for Family Research, University of Georgia, Athens, GA 30602, USA; 3Department of Psychological Sciences, University of Connecticut, Storrs, CT 06269, USA; 4Department of Sociology, University of Georgia, Athens, GA 30602, USA; 5Department of Psychiatry, University of Iowa, Iowa City, IA 52246, USA; 6Behavioral Diagnostics LLC, Coralville, IA 52241, USA

**Keywords:** alcohol, epigenetic aging, cg05575921, DNA methylation, immune cell function

## Abstract

Excessive alcohol consumption (EAC) has a generally accepted effect on morbidity and mortality, outcomes thought to be reflected in measures of epigenetic aging (EA). As the association of self-reported EAC with EA has not been consistent with these expectations, underscoring the need for readily employable non-self-report tools for accurately assessing and monitoring the contribution of EAC to accelerated EA, newly developed alcohol consumption DNA methylation indices, such as the Alcohol T Score (ATS) and Methyl DetectR (MDR), may be helpful. To test that hypothesis, we used these new indices along with the carbohydrate deficient transferrin (CDT), concurrent as well as past self-reports of EAC, and well-established measures of cigarette smoking to examine the relationship of EAC to both accelerated EA and immune cell counts in a cohort of 437 young Black American adults. We found that MDR, CDT, and ATS were intercorrelated, even after controlling for gender and cotinine effects. Correlations between EA and self-reported EAC were low or non-significant, replicating prior research, whereas correlations with non-self-report indices were significant and more substantial. Comparing non-self-report indices showed that the ATS predicted more than four times as much variance in EA, CDT4 cells and B-cells as for both the MDR and CDT, and better predicted indices of accelerated EA. We conclude that each of the non-self-report indices have differing predictive capacities with respect to key alcohol-related health outcomes, and that the ATS may be particularly useful for clinicians seeking to understand and prevent accelerated EA. The results also underscore the likelihood of substantial underestimates of problematic use when self-report is used and a reduction in correlations with EA and variance in cell-types.

## 1. Introduction

Despite decades of efforts, excessive alcohol consumption (EAC) continues to be one of the leading causes of morbidity and mortality in the United States and the world [1,2]. Alcohol exerts its effects on morbidity and mortality through two major pathways. First, chronic EAC leads to chronic medical conditions such liver cirrhosis, cancer, and cardiomyopathy that in turn lead to debilitating disease and death [3]. Second, acute EAC combined with potentially hazardous activities, especially driving, can cause lifelong disability or death not only for the drinkers, but for those around them [3]. All this excessive morbidity and mortality secondary to EAC is potentially preventable. For those not yet drinking heavily, preventive interventions for diminishing the likelihood of developing problem drinking are both readily available and effective [4,5]. For those already affected, evidence-based pharmacologic and psychotherapeutic treatments are also readily available [5,6]. Critically, because both preventive and active treatment inventions heavily rely on intact psychosocial support networks for the initiation and maintenance of healthy behaviors, prompt recognition of EAC is essential for obtaining truly effective clinical solutions. Hence, there is considerable desire from both the medical and public health community to develop methods to accurately detect and monitor EAC consumption.

For the acute detection of EAC, exhaled breath monitors or “breathalyzers” are both scalable and effective [7,8]. However, this testing solution is most commonly implemented in circumstances such as traffic accident and emergency room visits where the impact of EAC has already been realized and alcohol consumption is presumed to be quite recent. However, it is less useful in the routine clinical setting where the vast majority of healthcare encounters occur. As a consequence, the development of other methods through which EAC, particularly EAC characterized by periodic binge drinking, can be prospectively detected has been a continuing focus of the biomedical community. Developing methods to better detect and quantify the impact of binge drinking at the level of individual patient is a particularly important goal. Three-quarters of the total medical cost of alcohol misuse is thought to be related to binge drinking [9]. Better understanding of this pattern of excessive use may be particularly important in examining the impact of EAC for Black Americans, as EAC may be more stress-responsive among Black Americans, characterized by greater binge drinking but lower regular alcohol use [10,11,12]. Nonetheless, EAC among Black Americans has been associated with increased risk for adverse diseases of aging, such as CVD [13,14].

Traditionally, one of the most frequently used metrics for detecting any type of EAC has been changes in serum liver enzymes [7,8]. However, although still often used clinically, these methods are generally regarded as both insensitive and non-specific. As a result, over the past four decades, the biomedical research community has developed new purpose-driven biomarkers to quantify EAC.

The most successful of these other biomarkers are ethyl glucuronide (EtG), phosphatidyl ethanolamine (Peth) and carbohydrate deficient transferrin (CDT) [7,8,15,16,17]. EtG is produced as a minor metabolite of alcohol [8,18]. Although it can be assessed in serum, hair and other substrates, EtG quantification is generally conducted using urine, where it can be detected for up to 5 days after heavy consumption [19]. PEth is a cell membrane phospholipid that is produced from phosphatidylcholine by an enzymatically catalyzed transphosphatidylation reaction in the presence of ethanol [8]. Because the three-carbon glycerol backbone of the phospholipid accommodates two fatty acids in addition to its phosphate head to which the ethanolamine group is attached, and these two fatty acids can be attached to the 1, 2 or 1 and 3 carbons of the glycerol backbone, Peth actually is a family of phospholipids whose exact composition depends on the type and position of the fatty acid side chains. Depending on the exact species being detected, the serum half-life of Peth varies, with the most commonly detected variants having a serum half-life of 7 days. However, the assessment of PEth requires complex analytical procedures (i.e., gas or liquid chromatography), and samples must be frozen at −80 °C to prevent false positives and negative outcomes [15]. The third biomarker is carbohydrate-deficient transferrin (CDT) [16]. Transferrin is a serum iron binding protein that is post-translationally modified by the addition of two complex carbohydrate chains consisting of n-acetylglucosamine, mannose, galactose and sialic acid monomers. Increasing alcohol consumption is associated with lower rates or “deficiency” of sialic acid incorporation. Because sialic acid is a charged group, this allows for quantification of the amount of CDT by isoelectric electrophoretic techniques. Notably, the CDT has the longest half-life of the three biomarkers, with levels decaying back to normal approximately 2–3 weeks after abstinence [16]. Although each of these three biomarkers has a unique value in assessing EAC, as a result of a more favorable balance of sensitivity, specificity, half-life and ease of conduct, the CDT is generally considered the standard for alcohol testing with sensitivity and specificity of approximately 70% for heavy alcohol consumption [16].

A challenge to the more effective use of these purpose-driven alcohol tests is their limited breadth for guiding assessment and treatment. In general, clinical assessment tools tend to fall into one of two “buckets”. Measures, such as body mass index (BMI) and C-reactive protein (CRP), broadly predict risk and are usually easily obtained, but they often lack precision to diagnose or monitor treatment. In contrast, specialty assessments such as the CDT are specific for a given condition but require clinical suspicion before they are usually ordered. As a result, for routine medical assessments, clinicians tend to order broad based tools first, only adding more purpose driven assessments when the need is indicated.

Over the past several years, DNA methylation epigenetic aging (EA) indices have emerged as popular metrics for the assessment of chronic medical illness among wellness practitioners. Based on the original insights of Fraga and Esteller [20], and the more effective implementation strategies of Hannum and Horvath [21,22], these tools use complex regression algorithms to interpret the results from epigenome wide methylation arrays to infer overall health status using a metric termed “accelerated aging”, which is defined as the residual of EA controlling for the chronological age of the subject. In general, positive scores indicate that the person’s EA is greater than would be expected given their chronological age and are associated with the presence of medical disorders. Accelerated EA for some, but not all, EA indices have been predicted from non-self-report indicators of substance use [23,24,25,26,27].

With respect to alcohol consumption, the first widely used subscale the 144-probe based DNAm-Alc tool developed by Lui and colleagues in 2016, was initially reported to account for 15% of the variance in alcohol consumption. However, follow-up examinations by others using the DNAm-Alc metric found lower amounts of explained variance with one group speculating that the initial model developed by Lui and colleagues may have been overfitted [28,29]. Subsequent improvements in our understanding of best practices for generating predictive models has recently led to the development of a metric by Hillary and associates referred to as Methyl DetectR (MDR) [30]. MDR used a machine learning approach and data from 4450 participants in the Generation Scotland Study [31] to predict weekly alcohol consumption. However, neither of these two scales have been compared to established biomarkers of alcohol consumption, such as the CDT, or were tested in non-European subject populations. Hence, there is a pressing need for DNA methylation-based tools applicable to patients of all ancestries that exactly define discrete, targetable conditions for preventative and therapeutic medical interventions.

Using a slightly different, disease-focused approach, we have developed a methylation-sensitive digital PCR (MSdPCR) method for predicting heavy alcohol consumption (HAC), which is defined as drinking more than six drinks per day, which may be better for understanding the role of EAC to accelerated EA in patients and research subjects of all ancestries [32,33]. In direct head-to-head, blinded testing, this method, which uses the results from four MSdPCR assays to form an Alcohol T Score (ATS), significantly outperformed the CDT in predicting HAC status in a group of 313 subjects (182 controls, 131 HAC subjects) [33]. Furthermore, we have recently shown that in young Black American adults, variations of the ATS and another MSdPCR assessment of smoking account for 95% of all common variance in accelerated aging [34]. However, the relationship of the CDT to the ATS in non-treatment populations and their relative merits in predicting accelerated aging has not been established.

To better understand the utility of the MDR, ATS and CDT in the explication of accelerated EA in Black Americans, we now directly compare these measures to each other and to self-report with respect to accelerated aging as assessed by seven commonly used EA indices. In addition, we examine changes in self-report across two waves to better understand the impact of age-related reductions in self-reported problematic alcohol use and the impact on correlations with non-self-report indices of EAC. We also examine changes in correlations for smoking and drinking as a window on likely changes in the validity of self-report across waves.

## 2. Materials and Methods

The methods used to collect the clinical data and biomaterials from Wave 5 of the Family and Community Health Study, Target cohort (FACHS-T) were approved by the University of Iowa (IRB 201901770), Georgia and Connecticut Institutional Review Boards (IRB). These Wave 7 clinical interviews and blood draws were timed to occur within one week of one another in 2015 through 2016. We also utilize clinical interview data from Wave 6, which occurred in 2008–2009.

### 2.1. Clinical Data

The design and procedures used in this longitudinal study of health behaviors in Black American families have been described previously [35]. In brief, at Waves 6 and 7, after obtaining consent, each subject was interviewed over the phone with a structured interview that reviews key stressors and health variables. At Wave 7, research subjects were phlebotomized to provide biomaterial for laboratory analyses. During the clinical interview, smoking status was determined by asking the question “how many cigarettes have you smoked in the last 3 months?”. Those answering none were coded as non-smokers. HAC was determined by asking the question “During the past 12 months, how often have you had a lot to drink—that is 3 or more drinks at one time?”. Those answering 1 or more times per week were classified as unhealthy drinkers.

As part of our efforts to provide a non-coercive interviewing experience, subjects were not required to answer all questions and were instructed to skip questions that they did not feel comfortable answering. In addition, DNA or serum specimens may have been unavailable for certain subjects. As result, the total number of responses or assessments for any given question or assay may be less than the total number of subjects.

### 2.2. Molecular Data

Within one week of their clinical interview, each of the subjects was phlebotomized to provide biomaterial for these studies. After processing into DNA and serum via our usual methods, the samples were stored at −20 and −80 °C, respectively [36].

DNA Methylation: Genome-wide DNA methylation assessments using the Infinium MethylationEpic Beadchip (Illumina, San Diego, CA, USA) were conducted by the University of Minnesota Genome Center (http://genomics.umn.edu/ (accessed on 14 October 2022)) according to the manufacturer’s protocol. The resulting data were DASEN-normalized using the MethyLumi [37], WateRmelon [38], and IlluminaHumanMethylationEPICanno.ilm10b2.hg19 [39] R packages as per our previous descriptions [40]. Sample and probe level quality control of the data were then conducted as previously described [40]. In brief, samples were removed if more than 1% of their probes had detection *p* values of >0.05. After all processes were complete, values from 858,924 of the 866,091 probes in the array were retained.

β values for each site were calculated using the standard formula where U and M are the values of the unmethylated and methylated intensity probes (averaged over bead replicates), and α = 100 is a correction term to regularize probes with low total signal intensity [41,42]. CpG values were background-corrected using the “noob” method β = M/(U + M + α) [43].

This study features the use of 7 Epigenetic Aging indices developed over the past decade. The Hannum index was described in 2013 and consists of 71 CpG probes [21]. The Horvath Index, which like the Hannum index, was designed to calculate chronological age, was also first described in 2013 and consists of 353 CpG markers [22]. The PhenoAge index was initially described in 2018 and contains 513 markers. As opposed to the prior indices, it was developed using both clinical measures and age in an attempt to better predict individual differences in morbidity [44]. In 2019, another Horvath-led group introduced GrimAge, which uses the input from 1030 CpG probes to forecast all-cause mortality [45]. In a separate work published in 2019, the Horvath group also introduced the Telomere algorithm, which uses the input from 140 CpG sites to estimate telomere length in kilobases (Kb), which, in turn, is associated with aging [46]. Finally, using data from the Dunedin Multidisciplinary Health and Development Study, a group of investigators introduced first the DunedinPOAM (2020), and then the DunedinPACE (2022) algorithms [47,48]. The DunedinPOAM, which consists of 46 CpG probes, and DunedinPACE, which includes 173 CpG sites, reportedly provide a “speedometer” of aging designed to reflect physiological change over the past year, with values greater than one indicating accelerated biological aging. The values for the Hannum, Horvath, Levine, GrimAge, and Telomere metrics were calculated using the publicly available online tool hosted by the Horvath Lab (https://dnamage.genetics.ucla.edu/ (accessed on 1 July 2022)). The values for the DunedinPOAM and the DunedinPACE indices were calculated using the code supplied by the developers of the indices that is freely available at https://github.com/danbelsky (accessed on 1 July 2022). Epigenetic age acceleration for Horvath, Hannum, PhenoAge and GrimAge indices was calculated by using the unstandardized residual scores from the regression of epigenetic age on chronological age.

Cell-type composition was estimated using the “EstimateCellCounts” function in the “minfi” Bioconductor package, which is based on the method developed by Houseman and colleagues [49]. Using this approach, the white blood cell-type proportions (CD4+ T cells, Natural Killer cells and B cells) in whole blood specimens used to prepare the DNA for were estimated. Methyl DetectR values for alcohol consumption per week were calculated using the code supplied by the University of Edinburgh website (https://www.ed.ac.uk/centre-genomic-medicine/research-groups/marioni-group/methyldetectr (accessed on 1 July 2022)) [30].

Reference-free methylation sensitive digital PCR (MSdPCR) assessments of cg05575921 methylation, a generally accepted biomarker of smoking intensity, and Alcohol T Score (ATS), a recently introduced measure of Heavy Alcohol Consumption (HAC) were conducted using the same samples of DNA used in the conduct of genome-wide DNA methylation analyses [32,33,50,51]. The determination of methylation status at cg05575921 and the four loci (cg02583484, cg04987734, cg09935388 and cg04583842) used to form the Alcohol T Score (ATS) was conducted using fluorescent primer probe sets from Behavioral Diagnostics (Coralville, IA, USA) and both droplet digital PCR equipment and reagents from Bio-Rad (Hercules, CA, USA) as previously described [36,51]. The ATS is the sum of z-scores of four loci named above and is a zero-centered metric in abstinent populations [32,33]. Increasing amounts of alcohol consumption are positively associated with ATS levels with ATS values of 3.5 and 5 being suggestive and predictive of HAC (6 or more drinks per day) [33,52]. In contrast, the methylation assessments of cg05575921 are expressed as “% methylation”. Lifetime non-smokers have an average cg05575921 value of 86.6% ± 2.9 with levels of <80% being strongly predictive of smoking [51]. Increasing values of the ATS are predictive of increasing alcohol consumption with ATS values of 3.5 and 5 being suggestive and predictive of HAC (6 or more drinks per day) [33,52].

Data were analyzed using IBM SPSS statistics for Windows, Version 27.0. All reported R2 values are adjusted for the number of predictors and sample size. The descriptive statistics and frequency were used to describe the demographic and physiologic of self-report data by gender (see Table 1) and to depict the indices of epigenetic aging and alcohol use at Wave 7 (see Table 2). We then examined simple correlations and characterized the data in terms of mean and SD for study variables (see Table 3). In addition, the partial correlation was used to examine each of the metrics for alcohol consumption risk while controlling for gender and nicotine (see Table 4). Moreover, we used Pearson’s correlation analysis to examine the correlation between each of the metrics for alcohol consumption risk and the seven indices of epigenetic aging (See Table 5) and to assess the correlation between each of the metrics for alcohol consumption risk and cell types (see Table 6). Furthermore, a histogram was used to illustrate the normal distribution of CDT, MDR, and ATS (see Figure 1). 

## 3. Results

The demographic and substance use data relevant to the current investigation are given in Table 1. In brief, the cohort comprised Black Americans, majority female (62%), who averaged 28.7 ± 0.8 years of age at Wave 7 when all blood samples were drawn. They averaged 23.5 ± 0.8 years of age when clinical interviews were conducted at Wave 6. Overall, 31% of the cohort at Wave 6 and 20% of the cohort at Wave 7 (23% of males and 19% of females) self-reported smoking at least one cigarette in the three months prior to the interview. Unhealthy alcohol consumption was quantified by asking how many times an individual had consumed three or more drinks in one sitting. Only 37% of the cohort used in the current analyses (45% of male and 56% females) denied binge drinking at any time in the prior year at Wave 6, whereas 51.5% did so at Wave 7. Notably, at Wave 6, 46 participants (23 males and 23 females) reported unhealthy drinking one or more times per week, but at Wave 7, 18 participants reported unhealthy drinking one or more times per week. Accordingly, there was substantial self-reported desistence from Wave 6 to Wave 7, with many who reported heavy drinking at Wave 6 reporting little or no drinking at Wave 7.

Table 2 lists the biological markers of EA and substance consumption. Combustible tobacco consumption was assessed using DNA methylation at cg05575921. Overall, 41% of females (111 of 270) and 63% of males (106 of 167) had cg05575921 methylation levels of 80% or less indicative of current or recent daily smoking prior to assessment. In addition, tobacco consumption was further assessed using serum cotinine measures. Serum cotinine levels tended to be higher in males than females (*p* < 0.008 ANOVA). Alcohol consumption was assessed using the ATS, MDR and the CDT. CDT and MDR (*p* < 0.0001 ANOVA) but not ATS levels (N.S) were higher in male subjects than in female subjects. Figure 1 illustrates the overall distributions of CDT, MDR and ATS levels in the subjects. The CDT ranges from 0.1 to 11.7% with a marked right skew. The MDR ranges from −10.8 to −13.3 with no noticeable skewing. The ATS, which has a zero-centered distribution in abstinent individuals, ranges from −5.7 to 16.5 with a right skewing of the distribution.

Table 3 illustrates the relationship of the nine assessments of either smoking or alcohol consumption to one another. Self-reported smoking and drinking at Wave 6 were correlated with both self-reported smoking and drinking at Waves 7, as well as objective markers of smoking and drinking at Waves 7. Self-reported cigarette consumption at Wave 7 was correlated with objective markers of smoking and drinking, but self-reported drinking at Wave 7 was only correlated with objective indices of smoking, and MDR, not with other non-self-report indices of EAC, and observed significant correlations were attenuated from those observed with self-reported drinking from Wave 6. The correlation of cotinine with self-reported smoking was constant across Waves 6 and Wave 7 (r = 0.51, *p* < 0.001; and r = 0.54, *p* < 0.001 respectively). However, the correlation of cotinine with self-reported drinking at Wave 6 was r = 0.23, *p* < 0.001, but dropped to r = 0.14, *p* < 0.006, at Wave 7. The objective markers of smoking, cotinine and cg05575921, were strongly correlated with each other (r = −0.63, *p* < 0.001), and significantly correlated with all three non-self-report measures of alcohol use (absolute r’s from 0.16 to 0.58, all p’s < 0.01, overall mean absolute correlation = 0.34), and more strongly correlated with the non-self-report indices of EAC than they were with self-reported EAC at Wave 7 (absolute r’s = 0.14 and 0.15, overall mean absolute correlation = 0.145). The three objective alcohol markers were also moderately correlated with one another with correlations ranging from 0.37 to 0.41 (all *p* < 0.001).

Table 4 shows the effect of controlling for sex and presence of nicotine (yes vs. no) on the correlations between self-report and non-self-report indicators of alcohol use. As can be seen, the correlation of self-report at Wave 6 with non-self-report indicators (r’s = 0.124 to 0.189), as well as the intercorrelation of the non-self-report indicators (r’s = 0.363 to 0.365), are significant and robust with regard to these controls. Wave 7 self-reported EAC is not a significant predictor of any of the non-self-report indicators of EAC after controlling for potential confounding by sex and nicotine status.

Table 5 lists the correlations between the measure of EAC and EA. Self-reported problematic alcohol use is correlated poorly with EA across the board, although there is a positive correlation of SR6 with DunedinPOAM (r = 0.17, *p* < 0.001) and with GrimAge (r = 0.16, *p* < 0.001), as well as a negative correlation with telomere length (r = −0.11, *p* < 0.022), all in the expected direction. The CDT correlated modestly well with EA, in particular with POAM (r = 0.26, *p* < 0.001) and GrimAge (r = 0.32, *p* < 0.001), but also with telomere length (r = −0.14, *p* < 0.004), Horvath (r = −0.11, *p* < 0.002), and Hannum (r = 0.15, *p* < 0.002), with the average variance (R2) explained with respect to the seven indices being 3%. The array based MDR methylation metric performed similarly, with the moderate relationships observed with POAM (r = 0.19, *p* < 0.001), telomere length (r = −0.29, *p* < 0.001), and GrimAge (r = 0.31, *p* < 0.001), also explaining an average of 4% of the variance in EA. Finally, the digital methylation ATS measure had stronger relationships with EA indices, again particularly with respect to POAM (r = 0.58, *p* < 0.001) and GrimAge (r = 0.64, *p* < 0.001), but also with all other EA indices except Horvath, and overall explained an average of 18% of the variance in EA.

Finally, since alcohol is well known to have effects on the cells of the adaptive immune system [53], we examined the relationship of each of the measures of alcohol use on CD4T, natural killer (NK) and B lymphocyte levels, as quantified by the methylation array (see Table 6). Both the CDT and MDR were modestly positively correlated with NK (r’s = 0.11 and 0.16) but negatively correlated with B cell levels (r’s = −0.14 and −0.16). Finally, the ATS had a strong negative relationship with both CD4T (r = −0.31, *p* < 0.001) and B cell levels (r = −0.37, *p* < 0.001), but no relationship with NK cell count. Self-reported EAC at Wave 6 was unrelated to cells counts. Self-reported EAC at Wave 7 was associated with CD4T levels (r = 0.16, *p* < 0.001), but in the opposite direction expected.

## 4. Discussion

Longitudinal studies are critical resources for formulating a comprehensive chronologically informed understanding of the role of healthcare behaviors on the development of aging-associated diseases such as diabetes and cardiovascular disease. However, to be useful, the information derived from these studies must be both relevant and reliable. In this communication, we compare and contrast the association of three non-self-report metrics of alcohol consumption with epigenetic aging to the association observed using self-reported alcohol consumption. We found that while all three non-self-report indices of Excessive Alcohol Consumption (EAC): CDT, MDR, and ATS, correlated well with each other, with current and past wave indices of smoking, and with Wave 6 self-reported EAC, none of them correlated well with Wave 7 self-reported EAC. This is particularly interesting given that between Wave 6 and Wave 7, there was a marked shift toward decreased self-reported EAC among both males and females, a pattern typically taken as indicating “maturing out”. In addition, although non-self-reported indices of smoking correlated robustly with non-self-reported indices of EAC, as expected, given the well-known co-morbidity of cigarette use and EAC, and they were correlated with Wave 6 self-reported alcohol use, their association with Wave 7 self-reported EAC was attenuated. Together, these observations suggest either that there are strong lingering effects of Wave 6 EAC on the blood-based, non-self-report measures collected at Wave 7—a result that is biologically unlikely—or that the rapid decrease in self-reported EAC between Waves 6 and 7 reflects a greater change in self-report than in actual behavior. That is, “aging out” may be, in part, a function of changes in willingness to report problematic patterns of alcohol use.

We also explored the utility of non-self-report indices of EAC in the prediction of EA and key immune cell levels likely to be affected by EAC. Although self-reported alcohol use was unrelated to indices of EA, replicating prior findings showing modest to no effects of self-reported smoking and drinking on accelerated aging and mortality [54], we found correlations between non-self-report indices of EAC and EA, with all non-self-report indices of EAC predicting some measures of EA, and ATS outperforming CDT and MDR on average. Limitations of these findings include that this a single time point examination of an all Black American, young-adult cohort. It is possible that there are developmental changes in this group occurring between ages 23 and 29 that resulted in decreased validity of their self-reported substance use. For example, it may be that the transition into adulthood increased their concern about stigma associated with EAC. It is also possible that non-self-report indices may be responsive to a variety of contextual variables, perhaps functioning somewhat differently in different subpopulations. Nonetheless, the results help underscore the importance of incorporating non-self-report indices of EAC in the examination of health effects of elevated alcohol use.

Given the differences in the manner through which each of these measures of alcohol consumption were derived, some differences in their predictive capacities were to be expected. First, it is important to remember that the CDT and the two methylation metrics tap differing biological pathways. Alterations in serum CDT levels reflect changes in sialyltransferase activity in the liver [16,55]. In contrast, both the MDR and ATS measure methylation of DNA of cells from the hematopoietic system. Second, both measures with hypothesized detection windows, the CDT and the ATS, have differing half-lives. The CDT is expected to capture alcohol consumption over the prior three weeks [16]. In contrast, the half-life of the ATS appears to be at least several months [32]. Finally, the MDR can best be conceptualized as a continuous marker of alcohol consumption in the general population rather than an index of problematic use, as it was developed using average weekly alcohol consumption of Scottish subjects “who reported that their intake was representative of a normal week” [30]. In contrast, both the CDT and ATS were developed to detect those with alcohol use disorders. The CDT was initially developed during the last century using samples of cerebrospinal fluid from heavy drinkers affected by delirium tremens [56]. The ATS was developed quite recently using a case and control paradigm that contrasted the methylation of abstinent individuals with that of subjects who were hospitalized for alcohol intoxication in the context of at least 8 weeks of drinking at least 8 drinks per day [32]. As a consequence of all of these differences, it not surprising to find that the intercorrelations among the non-self-report indicators are relatively modest (0.36 to 0.41) and that they showed somewhat different distributions and patterns of association with indices of EA and blood-cell types. In contrast, the association of self-reported EAC with EA and white blood cell counts is extremely modest and quite different than that observed for the non-self-report indices of EAC.

The ATS has markedly stronger associations with EA and blood cell types than other non-self-report indicators of EAC. Because the ATS correlates better with EA indices and can be calculated from whole blood or saliva DNA, it may be a better tool, in many cases, than the CDT for those seeking to identify predictors of EA indices. However, because the array-based assessments of methylation at the four loci used in the ATS have poor precision, it would be necessary to conduct reference-free MSdPCR assessments as we did in the current investigation in order to use the ATS [32]. That is, computing the ATS from array-based assessments will not yield the same value as the reference free MSdPCR assessment, and cannot be used as a substitute. The MDR has the advantage over both the ATS and the CDT that it can be derived from existing array-based datasets without any additional assessment.

Despite the fact that the metrics used the same biological material, the ATS predicts EA much better than the MDR (i.e., over four times as much variance). This appears to reflect the fact that the ATS was designed to predict a damaging level of alcohol consumption (i.e., HAC), whereas the MDR was designed to predict weekly alcohol consumption. Since POAM and GrimAge were designed to predict morbidity and mortality, it seems logical that the ATS would predict EA better. Whether that extends to predicting lower levels of daily consumption in non-binge drinkers remains to be tested, and we note that only MethDectectR has been calibrated to accomplish this task.

Although we routinely use the ATS for both academic and commercial purposes, and believe we have a good understanding of its clinical predictive properties, we did not anticipate the strong relationship of the ATS, and the more modest correlation of the CDT and MDR, with both CDT T4 and B lymphocytes. In a prior analysis using these data, we have shown that the cell counts do not significantly affect the correlation between the EA indices or the ability of the ATS and cg05575921 to predict the common variance of these indices [34]. Nevertheless, since EAC has well-known effects on immune system function [53], we conducted the exploratory analyses detailed in Table 5. Increasing levels of alcohol consumption were strongly negatively correlated with levels of CDT4 and B lymphocytes. These findings agree well with our prior understanding of alcohol consumption on immune function [53,57]. In contrast, it is difficult to reconcile the positive associations of the CDT and MDR with natural killer (NK) cells, or the positive association of self-reported alcohol use with CDT4. Most studies of alcohol consumption show that increasing amounts of alcohol consumption are associated with decreasing levels of NK cells [58,59,60]. Still, we note that correlations of the CDT4 and MDR with NK cell count are modest. Nevertheless, whether any of these metrics predict future disease associated with impaired immune cell function, such as chronic bronchitis, in this population, is unknown, and alcohol consumption has both pro- and anti-inflammatory effects whose exact impact is disease- and dose-dependent [57,61,62]. However, we believe that these data support the future use of ATS and these cell count measures for developing a more granular understanding of the relationship of alcohol consumption to inflammatory-related illness in longitudinal populations with DNA biobanks.

Of particular note is the poor predictive value of self-reported drinking at Wave 7 when the participants were 29 years of age. Because of the work of ourselves and others demonstrating the often-poor reliability of self-reported smoking in high risk and underserved populations, as well as in White, non-high risk populations [52], and the generally acknowledged stigma against being a chronic abuser of alcohol, we are not overly surprised [34,63,64,65]. It is quite possible that as young adults assume greater responsibility, they become more responsive to alcohol-related stigma, perhaps decreasing EAC somewhat, but perhaps decreasing their self-report more. Regardless of the various sets of reasons for underreporting that may emerge in different groups, these results add further emphasis for the need to incorporate biomarkers of smoking and drinking in all studies of accelerated aging and diseases of aging when possible. Furthermore, we believe that the strong relationships of the ATS and cg05575921—now shown in three populations [52,66]—may suggest the need to revisit prior findings that suggested a low correlation between tobacco use and drinking [67].

At the same time, the current results suggest an opportunity provided by the use of non-self-report indices of EAC to identify subjects who are likely providing unreliable self-report, or who have recently changed from providing more accurate to less accurate self-report information about their EAC. Using this opportunity to better characterize such individuals and better understand the likely pressures leading to substantial underreporting could lead to opportunities to make substantial improvements in both research and clinical care. To a certain extent, no matter what we do, social stigma against excessive alcoholism and other similar behaviors will continue to exist, and we are not in favor of policies that lead to unwanted intrusions into privacy. Still, by better defining the characteristics of those likely to provide unreliable self-report, we may be able to formulate effective adjustments to our approaches for gathering the necessary information from these individuals for both research and clinical purposes thus creating a societal benefit. In addition, through careful analyses of these and other datasets, it may be possible to identify the scope of associated unreliable self-reporting and mitigate its effects on analyses of similar health related outcomes.

## 5. Conclusions

In summary, we conclude that the ATS better predicts EA and alcohol-related cell count changes than either the MDR or CDT. However, the three non-self-report indices all had unique strengths and represent very different windows on EAC. Greater attention to the use of some non-self-report index of EAC seems important in future research on the health impact of EAC, and the use of any of these biomarkers in studies of aging-associated disease may improve the strength and reliability of any subsequent findings.

## Figures and Tables

**Figure 1 genes-13-01888-f001:**
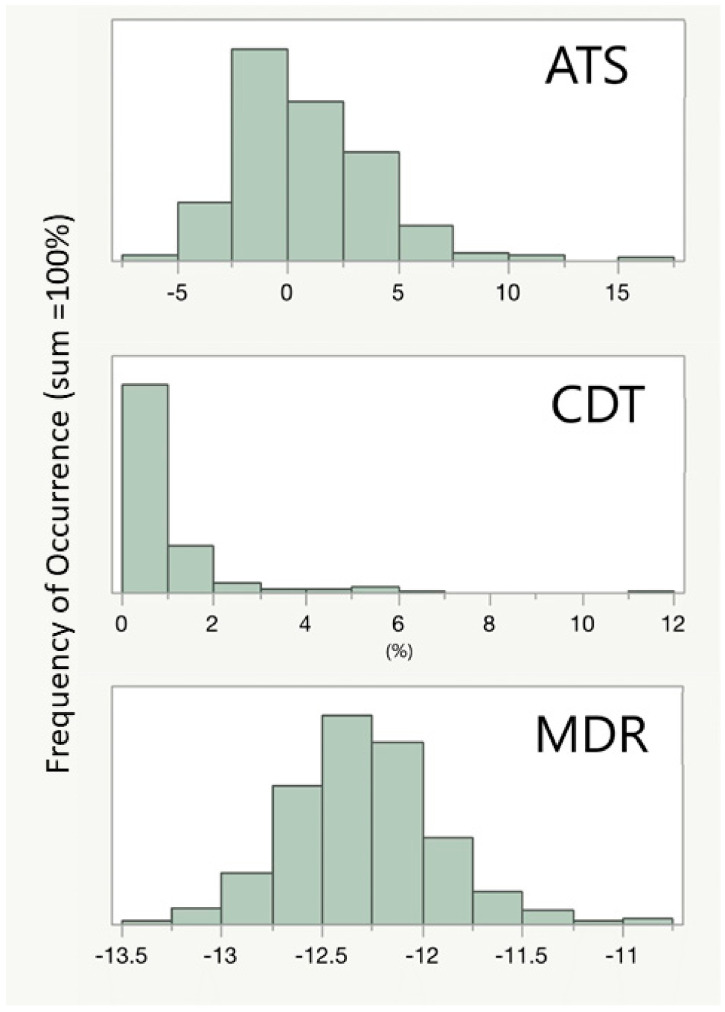
The distribution of the ATS, CDT and the MDR in the FACHS T population. Both the ATS and the MDR are unitless. The CDT is expressed as percent of total transferrin.

**Table 1 genes-13-01888-t001:** Wave 6 and Wave 7 demographic, physiologic and self-report data.

Variable	Wave 6 (2008–2009; *n* = 437)		Wave 7 (2015–2016; *n* = 437)	
	Male	Female	Male	Female
N	167	270	167	270
Mean Age	23.46 ± 0.9	23.5 ± 0.8	28.6 ± 0.8	28.7 ± 0.8
Self-reported Smoking				
Yes	55	81	38	51
No	102	185	110	202
No Answer	10	4	19	17
Self-reported non-combustible nicotine use				
None	N/A	N/A	163	267
1–5 times	N/A	N/A	2	3
>5 times	N/A	N/A	2	0
Self-reported Unhealthy Alcohol Use				
Never	47	115	75	150
1–2 times	32	60	35	68
About 3–11 times	19	28	20	27
A few times per month	26	35	9	12
About 1–2 times per week	14	15	6	4
Several times per week	9	8	5	3
Don’t know	3	2	8	1
Refused	17	7	9	5

**Table 2 genes-13-01888-t002:** Indices of Epigenetic Aging and Alcohol Use at Wave 7.

	Male	Female
N	167	270
Smoking		
Cg05575921	66% ± 19	75% ± 16
<80%	106	111
≥80%	59	158
ELISA		
Cotinine > 2 ng/ml		
Yes	98	115
No	54	142
Cotinine (ng/mL)	57 ± 63	41 ± 58
Drinking		
ATS	1.04 ± 3.3	0.60 ± 2.7
MDR	−12.17 ± 0.4	−12.38 ± 0.3
CDT	1.22% ± 1.40	0.77% ± 0.37
Accelerated Aging		
Dunedin PACE7	0.97 ± 0.11	1.03 ± 0.13
Dunedin POAM7	1.08 ± 0.09	1.08 ± 0.09
Horvath	0.88 ± 4.33	−0.54 ± 4.13
Hannum	0.92 ± 4.13	−0.56 ± 3.59
PhenoAge	−1.20 ± 5.89	0.74 ± 6.75
GrimAge	1.12 ± 4.65	−0.68 ± 4.12
Telomere Age	7.57 ± 0.21	7.66 ± 0.16

Note: ATS, Alcohol T Score; CDT, carbohydrate deficient transferrin; MDR = Methyl DetectR.

**Table 3 genes-13-01888-t003:** Pearson correlation coefficients of each of the metrics for tobacco or alcohol consumption risk to themselves.

	1	2	3	4	5	6	7	8	9
1. SR Smoking6	—								
2. SR Smoking7	0.59 **	—							
3. SR6 Alcohol	0.23 **	0.20 **	—						
4. SR7 Alcohol	0.14 **	0.17 **	0.39 **	—					
5. Cotinine	0.51 **	0.54 **	0.23 **	0.14 **	—				
6. cg05575921	−0.51 **	−0.45 **	−0.21 **	−0.15 **	−0.63 **	—			
7. ATS	0.33 **	0.19 **	0.20 **	0.02	0.36 **	−0.58 **	—		
8. CDT	0.16 **	0.15 **	0.16 **	0.08 ^†^	0.16 **	−0.31 **	0.37 **	—	
9. MDR	0.24 **	0.16 **	0.17 **	0.12 *	0.24 **	−0.39 **	0.37 **	0.41 **	—
Mean	0.32	0.22	10.45	0.93	0.52	710.78	0.77	0.94	−120.30
SD	0.46	0.42	10.58	10.37	0.50	170.71	20.92	0.94	0.36

*p* < 0.1, * *p* < 0.05, ** *p* < 0.01, ^†^
*p* < 0.1. Note: ATS, Alcohol T Score; CDT, carbohydrate-deficient transferrin; MDR = Methyl DetectR.

**Table 4 genes-13-01888-t004:** Partial correlation coefficients of each of the metrics for alcohol consumption risk to themselves after controlling sex and nicotine.

	SR6 Alcohol	SR7 Alcohol	ATS	CDT	MDR
SR6 Alcohol	—				
SR7 Alcohol	0.367 **	—			
ATS	0.189 **	0.006	—		
CDT	0.124 *	0.044	0.364 **	—	
MDR	0.131 **	0.070	0.363 **	0.365 **	—
Mean	10.452	0.931	0.767	0.939	−120.298
SD	10.578	10.374	20.923	0.939	0.364

* *p* < 0.05, ** *p* < 0.01. Note: ATS, Alcohol T Score; CDT, carbohydrate-deficient transferrin; MDR, Methyl DetectR; SD, standard deviation.

**Table 5 genes-13-01888-t005:** Pearson correlation coefficients and average variance explained for each of five metrics of alcohol consumption with respect to each of the seven indices of EA.

	PACE	POAM	Telomere	∆Horvath	∆Hannum	∆PhenoAge	∆GrimAge	Avg R^2^ Overall
SR6 Alcohol	0.03	0.17 **	−0.11 *	0.02	0.05	−0.02	0.16 **	0.01
SR7 Alcohol	−0.13 **	−0.01	0.04	−0.04	−0.05	−0.10	0.02	0.00
ATS	0.35 **	0.58 **	−0.49 **	0.03	0.26 **	0.33 **	0.64 **	0.18
CDT	0.09	0.26 **	−0.14 **	−0.11 *	0.15 **	0.08	0.32 **	0.03
MDR	−0.12 *	0.19 **	−0.29 **	0.00	0.07	0.01	0.31 **	0.04

* *p* < 0.05, ** *p* < 0.01. Note: ATS, Alcohol T Score; CDT, carbohydrate-deficient transferrin; MDR, Methyl DetectR.

**Table 6 genes-13-01888-t006:** Pearson correlation coefficients of alcohol consumption markers with predicted cell counts.

	CD4T	NK	B Cell
SR6 Alcohol	0.02	−0.02	−0.08
SR7 Alcohol	0.16 **	−0.08	0.07
ATS	−0.31 **	−0.00	−0.37 **
CDT	−0.09 ^†^	0.11 *	−0.14 **
MDR	−0.06	0.16 **	−0.16 **

* *p* < 0.05, ** *p* < 0.01, ^†^
*p* < 0.1.

## Data Availability

The datasets used during the current study are available from the corresponding author on reasonable request.

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
