# Peer review of "Epigenetic and Proteomic Biomarkers of Elevated Alcohol Use Predict Epigenetic Aging and Cell-Type variation Better Than Self-Report"

_genes, 2022, doi:10.3390/genes13101888_

Round 1

Reviewer 1 Report

Dear Authors,

In my opinion your article “Epigenetic and Proteomic Biomarkers of Elevated Alcohol Use Predict Epigenetic Aging and Cell-Type variation Better than Self-Report “is very interesting and well prepared, but I have suggestions considering small improvements, as listed below:

Line 15: “As” instead of  “but” would be better

Line 19 : “may be helpful” instead of “may help”

Line 43: “drinkers” not “drinker”

Line 50: Instead of using “ fostering” maybe “obtaining” would be better.

Line 88: Peth should be replaced by PEth

Author Response

Please see attachment that covers both sets of revisions.

Reviewer 2 Report

The following manuscript aimed to compare self-report indices of drinking behavior with biomarker approaches in a minority population. The paper does uncover some strengths and weaknesses for these methods, including the potential for under reporting. This is a decently written paper and certainly there is a need to address the validity of the self-report measures. The authors should, in turn, also discuss the validity of the biomarkers rather than assuming they are the true gold standard. Additionally, there are a few other points.

Major 

  1. More detail description of the statistical tests done is needed. Did they test for normality of data for linear assumption? The CDT measure does not look like a normal distribution
  2. The “predictive” language may be a bit strong here. Although in some cases the model is predicting an outcome variable, this appears to be more of an associative relationship.
  3. There discrepancy in wave 7 is a bit intriguing but also confusing. It is possible that there is under reporting. However, the authors should further explore other factors that could contribute to this. For example, what are the limitations with the ATS itself?
  4. It was hard to follow the meaning and significance of the different waves, please clarify or perhaps put this information in a figure/table?

Minor

  1. Figure 1 is missing the y axis label. 
  2. Spacing is inconsistent within the document
  3. It would be helpful to more clearly state the hypothesis to be tested.

Author Response

(The authors gave the same response as above.)

Round 2

Reviewer 2 Report

I have no further comments, and thank the authors for addressing previous concerns.